# Cross-Talk between CB_1_, AT_1_, AT_2_ and Mas Receptors Responsible for Blood Pressure Control in the Paraventricular Nucleus of Hypothalamus in Conscious Spontaneously Hypertensive Rats and Their Normotensive Controls

**DOI:** 10.3390/cells11091542

**Published:** 2022-05-04

**Authors:** Krzysztof Mińczuk, Eberhard Schlicker, Barbara Malinowska

**Affiliations:** 1Department of Experimental Physiology and Pathophysiology, Medical University of Białystok, ul. Mickiewicza 2A, 15-222 Białystok, Poland; krzysztof.minczuk@umb.edu.pl; 2Department of Pharmacology and Toxicology, University of Bonn, Venusberg-Campus 1, 53127 Bonn, Germany; e.schlicker@uni-bonn.de

**Keywords:** angiotensin II, angiotensin 1–7, AT_1_ receptor, AT_2_ receptor, blood pressure, cannabinoid CB_1_ receptor, CP55940, Mas receptor, paraventricular nucleus of hypothalamus, spontaneously hypertensive rats

## Abstract

We have previously shown that in urethane-anaesthetized rats, intravenous injection of the angiotensin II (Ang II) AT_1_ receptor antagonist losartan reversed the pressor effect of the cannabinoid CB_1_ receptor agonist CP55940 given in the paraventricular nucleus of hypothalamus (PVN). The aim of our study was to determine the potential interactions in the PVN between CB_1_ receptors and AT_1_ and AT_2_ receptors for Ang II and Mas receptors for Ang 1–7 in blood pressure regulation in conscious spontaneously hypertensive (SHR) and normotensive Wistar Kyoto (WKY) rats. The pressor effects of Ang II, Ang 1–7 and CP55940 microinjected into the PVN were stronger in SHRs than in WKYs. Increases in blood pressure in response to Ang II were strongly inhibited by antagonists of AT_1_ (losartan), AT_2_ (PD123319) and CB_1_ (AM251) receptors, to Ang 1–7 by a Mas antagonist (A-779) and AM251 and to CP55940 by losartan, PD123319 and A-779. Higher (AT_1_ and CB_1_) and lower (AT_2_ and Mas) receptor expression in the PVN of SHR compared to WKY may partially explain the above differences. In conclusion, blood pressure control in the PVN depends on the mutual interaction of CB_1_, AT_1_, AT_2_ and Mas receptors in conscious spontaneously hypertensive rats and their normotensive controls.

## 1. Introduction

Hypertension is one of the main risk factors for cardiovascular diseases, but in ~90–95% of the patients with so-called primary, essential or idiopathic hypertension, its specific etiology is unclear [1]. Genetic and environmental factors and alterations of peripheral and central sites involved in blood pressure (BP) regulation are being considered [2,3].

For the central regulation of cardiovascular functions, brain stem areas such as the rostral ventrolateral medulla (RVLM) and the nucleus tractus solitarius (NTS) and the hypothalamic paraventricular nucleus (PVN) play an important role [1,4]. PVN is a main source of stimulation for sympathetic outflow that is finely tuned by both sympathoexcitatory (glutamate) and sympathoinhibitory (γ-aminobutyric acid; GABA) transmitters. The hypothalamic balance of inhibitory and excitatory synaptic inputs is impaired in primary hypertension [1,4].

PVN contains all components of the renin-angiotensin system (RAS), including angiotensin II (Ang II), angiotensin 1–7 (Ang 1–7) and the receptors of Ang II (AT_1_ and AT_2_) and Ang 1–7 (Mas), which are key modulators in cardiovascular functions [5,6,7]. Ang II is regarded as one of the sympathoexcitatory transmitters in the PVN [1]. Two classes of RAS blockers, i.e., angiotensin-converting enzyme (ACE) inhibitors (e.g., enalapril) and AT_1_ receptor antagonists (e.g., losartan), serve as first-line drugs for the treatment of essential hypertension according to the guidelines of American [3] and European [8] heart societies.

The following facts confirm the involvement of Ang II in the PVN in the pathophysiology of hypertension: (1) In human PVN, the expression of prorenin in hypertension positively correlated with the level of blood pressure [9]; (2) Ang II microinjected into the PVN induced stronger pressor responses and/or increases in sympathetic activity in renovascular hypertensive rats than in their normotensive controls [10,11]; (3) a higher expression of AT_1_ receptors (AT_1_Rs) was found in the PVN of renovascular and spontaneously hypertensive rats (SHR) [11,12,13] than in normotensive animals and (4) BP decreased in SHR but not in Wistar rats in response to AT_1a_ receptor gene silencing [14].

Ang 1–7 is mainly known for its vasodilatory/hypotensive responses (for review, see [5,7]); however, its central administration (including intra-PVN injections) increases BP [5,11]. Pressor responses were enhanced, and concomitantly, a higher density of Mas receptors (MasRs) in PVN was observed in renovascular hypertensive rats compared to their normotensive controls [11,15]. The MasR antagonist A-779 microinjected into the PVN reduced BP and sympathetic activity in high salt hypertensive rats but not in their controls [16].

Cannabinoids such as phytocannabinoid Δ^9^-tetrahydrocannabinol (THC), synthetic cannabinoid CP55940 or endocannabinoid anandamide (AEA) exert complex effects on the cardiovascular system that are influenced by hypertension. Intravenous (i.v.) injection mainly decreases BP in anaesthetized rats but increases it in conscious animals (for review, see [17,18]). We have shown previously that microinjection of methanandamide (MethAEA; the stable analogue of AEA) and CP55940 into PVN led to a cannabinoid CB_1_ receptor (CB_1_R)-mediated hypotension in urethane-anaesthetized rats; this hypotension was, however, reversed into a pressor effect by the CB_1_R antagonist AM251 when administered via the i.v. route [19].

Two pieces of evidence confirm the existence of an interaction between Ang II and CB_1_Rs in the PVN of urethane-anaesthetized normotensive rats. Firstly, in our previous work [20], the AT_1_R antagonist losartan i.v. not only inhibited the pressor effect of CP55940 (plus AM251 i.v.) administered into PVN but even reversed it into a hypotensive one. Secondly, Gyombolai et al. [21] showed that AM251 reduced the increase in BP elicited by microinjections of Ang II (both compounds were given to the PVN). Moreover, the enhancement of CB_1_ and CB_2_ receptor expression in the rat uterus in response to local Ang 1–7 infusion [22] indicates a potential interaction between CB receptors and Ang 1–7.

The aim of our study on conscious rats was to determine the interactions in the PVN between cannabinoid CB_1_Rs with AT_1_ and AT_2_ receptors for Ang II and Mas receptors for Ang 1–7 involved in blood pressure regulation. SHR, which represent a model of essential hypertension in humans and in which most antihypertensive drugs are active [23], were compared to their normotensive controls—Wistar Kyoto rats (WKY).

## 2. Materials and Methods

### 2.1. Animals

Male spontaneously hypertensive rats (SHR) and Wistar Kyoto (WKY) rats, weighing at 250–300 g, were purchased from the Center for Experimental Medicine of the Medical University of Białystok (Poland). After an acclimatization period of three days, each SHR and WKY rat was assigned randomly to one of the experimental groups. All surgical procedures and experimental protocols were executed in accordance with European Directive (2010/63/EU) and Polish legislation and were approved by the Local Animal Ethics Committee in Olsztyn (Poland; approval code: 77/2019; approval date: 29 October 2019). The study was performed following the principles of replacement, refinement or reduction (the 3Rs). Rats were housed at constant humidity (55 ± 5%) and temperature (22 ± 1 °C) and were kept under a 12/12 h light/dark cycle. Animals had free access to standard pelleted rat chow and tap water.

### 2.2. Placement of Cannula for PVN Microinjection

Rats were anaesthetized intraperitoneally (i.p.) with pentobarbitone sodium (300 μmol/kg). To access PVN, each rat was placed in a stereotaxic frame (World Precision Instruments; Sarasota, FL, USA). The stereotaxic coordinates for PVN, according to the Paxinos and Watson rat brain atlas [24], were as follows: 1.8 mm caudal from bregma, 0.5 mm lateral (right) to the midline and 8.0 mm below the skull surface. Two 1 mm holes were drilled into the skull, one using the above-mentioned coordinates, for guide cannula implantation. The cannula had an outer (OD) and inner (ID) diameter of 0.5 and 0.3 mm, respectively, and a length of 7.5 mm (therefore, its lower end was 0.5 mm above the injection site, thus preventing potential damage of the PVN). The second hole was drilled few millimeters for placing a 1 mm metal screw for better fixing the cannula in place with dental cement. The cannula was then plugged with a 30 G dummy wire cannula.

### 2.3. Placement of Catheters for BP Measurements and Injections

Right after cannulation, under the same anesthesia, animals were placed on a heated table. One cm incisions were made between the scapulae and on the neck and groin area. Carotid artery and femoral vein were exposed and separated from adjacent tissues. Next, polyurethane catheters (1 mm OD and 0.63 mm ID), filled with heparinized saline, were inserted into carotid artery and femoral vein, and their extravascular parts were tunneled under the skin and exteriorized on the neck. Immediately after insertion, the catheter was flushed with heparinized saline (100 units/mL) in order to prevent the formation of a blood clot. Rats were then returned to their individual cages and allowed to recover for 3 days. During that time, catheters were flushed daily with heparinized saline to ensure their patency. During surgery, animals received buprenorphine (0.05 mg/kg subcutaneously) and all surgical sites were disinfected with betadine (100 mg/mL) before making any incisions. During the recovery period, rats were given paracetamol (100–300 mg/kg/day) in drinking water.

### 2.4. Microinjections

All PVN microinjections were performed on freely moving, conscious rats in a volume of 200 nl and completed in ten seconds. The injection cannula was longer by 0.5 mm than the guide cannula to make sure that the compounds were delivered to the correct site. When multiple injections were administered, the next injection was only given when BP had returned to the baseline level. At the end of the experiment, 100 nl of 2% Evans Blue was injected into the microinjection site for the histological identification of PVN with an optical microscope. Only animals with the microinjection site within the confines of the PVN were included in data analysis. A representative photo of the microinjection site evaluated by Evans Blue is shown in Figure 1.

### 2.5. Experimental Protocol

Each rat received two unilateral microinjections into the PVN (S1 and S2). The agonists Ang II, Ang 1–7 or CP55940 were administered into the PVN once (S2) at the following doses: 0.3 nmol/rat [10,11], 0.3 nmol/rat [11] and 0.1 nmol/rat [20], respectively. Antagonists of AT_1_ (losartan, 20 nmol/rat; [25]), AT_2_ (PD123319, 10 nmol/rat; [25]), Mas (A-779, 3 nmol/rat; [11]) and CB_1_ (AM251, 30 nmol/rat, [19]) receptors were injected into the PVN 5 min before the respective agonists (or after BP had returned to preinjection values; S1). Control groups received the respective vehicle for the particular receptor ligands. All experiments with CP55940 were performed in the presence of AM251 (3 μmol/kg) given i.v. (into the femoral vein in a volume of 0.5 mL/kg) in order to prevent CP55940-induced hypotension (according to [20]). Intravenous injection of AM251 induced a long-lasting pressor effect (for details, see Results) and the microinjection of the respective receptor antagonists into the PVN took place after the return of basal BP to preinjection values.

Cardiac parameters were recorded from the right carotid artery of unrestrained conscious rats by a pressure transducer system (model MLT844, ADInstruments, Sydney, Australia) with computer data acquisition (Bridge Amp/PowerLab 4/35, ADInstruments, Sydney, Australia). We did not analyze heart rate since the changes induced by the particular ligands were too small and showed a big scatter in variation.

### 2.6. PVN Tissue Microdissection

At the end of the experiment, rat brains were sectioned into 1 mm slices using a rat brain matrix (Zivic Instruments, Pittsburgh, PA, USA) at the appropriate coordinates taken from the Paxinos and Watson rat brain atlas [24] for PVN, RVLM and NTS. Tissues were then isolated by the use of a punch-out technique, flash-frozen in liquid nitrogen and stored at −80 °C until analysis by Western blotting.

### 2.7. Western Blot Analysis

Routine Western blotting procedures were used. Briefly, the samples of PVN, RVLM, and NTS from each rat were homogenized in ice-cold M-PER lysis buffer (Thermo Scientific, Rockford, IL, USA) containing a cocktail of protease inhibitors (Roche Diagnostics GmbH, Mannheim, Germany). Total protein concentration was determined using the bicinchoninic acid method (BCA) (Pierce Rapid Gold BCA Protein Assay Kit, Thermo Fisher Scientific, Waltham, MA, USA) with bovine serum albumin as a standard. Next, homogenates were reconstituted in Laemmli buffer with β-mercaptoethanol, separated by 10% sodium dodecyl sulfate-polyacrylamide gel electrophoresis and transferred onto nitrocellulose membranes. The membranes were incubated with the primary antibodies of interest (anti-CB_1_R, ab259323; anti-AT_2_R, ab92445; anti-MasR, ab200685; Abcam, Cambridge, UK) followed by incubation with appropriate secondary antibodies conjugated with horseradish peroxidase (ab6721, Abcam, Cambridge, UK). Protein bands were visualized using an enhanced chemiluminescence substrate (Thermo Scientific, Rockford, IL, USA) and quantified using ImageJ software (v1.53f51). Protein expression was standardized to β-actin or glyceraldehyde-3-phosphate dehydrogenase (GAPDH) depending on the molecular weight of the specific antibodies.

### 2.8. Data Analysis

Results are given as means ± SEM; n refers to the number of rats. In order to quantify the effects of antagonists on the cardiovascular effects of Ang II, Ang 1–7 or CP55940, the agonist-induced maximal change in BP was calculated from the respective basal systolic, diastolic and mean BP (SBP, DBP and MBP) averaged over 5 min before the injection of the particular agonist. This procedure was chosen to minimize the influence of inter-subject variability on final data. For a comparison of the mean values between the WKY and SHR group, the t-test for unpaired data was used. When two groups were compared with the same control, one-way analysis of variance (ANOVA) followed by the Dunnett post hoc test was used. Differences were considered as significant when *p* < 0.05. Statistical analysis was performed using Graph Pad Prism 5 (GraphPad Software, La Jolla, CA, USA).

### 2.9. Drugs

A-779 (5-L-isoleucine-7-D-alanine-1-7-angiotensin II, trifluoroacetate salt); AM251 [(N-(piperidin-1-yl)-5-(4-iodophenyl)-1-(2,4-dichlorophenyl)-4-methyl-1H-pyrazole-3-carboxamide)]; angiotensin 1–7; angiotensin II (Tocris Bioscience, Bristol, UK); betadine (Egis Pharmaceuticals PLC, Budapest, Hungary); buprenorphine (Richter Pharma AG, Wels, Austria); CP55940 [(−)-cis-3-[2-hydroxy4-(1,1-dimethylheptyl)phenyl]-trans-4-(3-hydroxypropyl)cyclohexanol] (Sigma Aldrich, St. Louis, MO, USA); losartan potassium (Tocris Bioscience, Bristol, UK); paracetamol (Sequoia, Warsaw, Poland); PD123319 ((6S)-1-[[4-(dimethylamino)-3-methylphenyl]methyl]5-(2,2-diphenylacetyl)-4,5,6,7-tetrahydro-1H-imidazo[4,5-c]pyridine-6-carboxylic acid, di(2,2,2-trifluoroacetate)) (Sigma Aldrich, St. Louis, MO, USA); pentobarbitone sodium (Biowet, Puławy, Poland).

Drugs were dissolved in saline with the following exceptions. AM251 was dissolved in a mixture of ethanol, Cremophor El, DMSO and saline (1:1:1:9.5) for i.v. injections and in DMSO and saline (1:9) for PVN injections. CP55940 was dissolved in a 19% solution of cyclodextrin. None of the used vehicles significantly altered any of the cardiovascular parameters by itself (data not shown).

## 3. Results

### 3.1. General

Basal systolic (SBP), diastolic (DBP) and mean (MBP) blood pressure, measured immediately before the administration of the antagonists (S_1_) into the PVN, was significantly higher in spontaneously hypertensive rats (200 ± 4, 168 ± 3 and 184 ± 3 mmHg, respectively; *n* = 80; *p* < 0.001) than in normotensive Wistar Kyoto rats (129 ± 2, 105 ± 3, 117 ± 2 mmHg, respectively; *n* = 80). The CB_1_R antagonist AM251 given i.v. (3 μmol/kg) and in PVN (30 nmol/rat) increased BP by about 15–20 mmHg and 5–10 mmHg, respectively (Figure 2a,b). AM251-induced pressor effects lasted for ~8–15 and ~5–7 min after its application i.v. and in PVN, respectively. The increases in BP elicited by i.v. AM251 injection were slightly higher in SHR in comparison to WKY although a significant difference was obtained for MBP only. AT_1_ receptor antagonist losartan 20 nmol/rat and AT_2_ receptor antagonist PD123319 10 nmol/rat, microinjected into the PVN did not modify basal BP by themselves (results not shown). Interestingly, as shown in Figure 2c,d, the MasR antagonist A-779 (3 nmol/rat) given in PVN did not have any effect on BP in the WKY or SHR group by itself. However, after the blockade of CB_1_Rs with AM251 i.v., it decreased SBP, DBP and MBP by about 5–10 mmHg both in WKY and in SHR.

### 3.2. Involvement of AT_1_, AT_2_ and Mas Receptors in the Pressor Responses of Ang II and Ang 1–7 Microinjected into PVN

The AT_1_ and AT_2_ receptor agonist Ang II 0.3 nmol/rat, the MasR agonist Ang 1–7 0.3 nmol/rat and the CB_1_R agonist CP55940 0.1 nmol/rat (given in the presence of AM251 3 μmol/kg, i.v.) microinjected into PVN-enhanced SBP, DBP and MBP both in WKY and in SHR (for original tracings, see Figure 3). In WKY, Ang II induced comparable increases in SBP, DBP and MBP by about 20 mmHg (Figure 4a). They were significantly higher in SHR by about 60%, 40% and 50%, respectively (Figure 4b). All pressor effects of Ang II both in WKY (Figure 4a) and in SHR (Figure 4b) were almost completely blocked by the previous injection of the AT_1_R antagonist losartan (20 nmol/rat) into PVN and inhibited by ~75% by the AT_2_R antagonist PD123319 (10 nmol/rat).

Compared to Ang II, Ang 1–7 increased BP to a lesser extent both in WKY and SHR (Figure 4c,d). In WKY, SBP, DBP and MBP increased by ~6 mmHg, while SHR showed significantly higher values in SBP (by ~200%) and MBP (by ~100%) compared to WKY. The increase in DBP (by ~50%) did not reach statistical significance. The pressor effects of Ang 1–7 both in WKY and in SHR were diminished by the previous injection of the MasR antagonist A-779 (20 nmol/rat) by ~75% (Figure 4c,d).

### 3.3. Involvement of CB_1_ Receptors in the Pressor Effects of Ang II and Ang 1–7 Microinjected into the PVN

As shown in Figure 5, the CB_1_R antagonist AM251 injected into the PVN before Ang II and Ang 1–7 injected also in PVN modulated their pressor responses. AM251 significantly lowered the increase in MBP induced by Ang II and tended to cause lower Ang II-induced increases in DBP and MBP in WKY (Figure 5a). In SHR, AM251 reduced Ang II-induced increases in SBP, DBP and MBP by 55%, 85% and 65%, respectively (Figure 5b). Interestingly enough, the pressor effects of Ang 1–7 (SBP, DBP and MBP) were not only reduced by AM251, but they even reversed into hypotensive responses both in WKY (Figure 5c) and SHR (Figure 5d).

### 3.4. Involvement of AT_1_, AT_2_ and Mas Receptors in the Pressor Effect of the CB_1_ Receptor Agonist CP55940 Microinjected into the PVN

Like in our previous study [20], CP55940 increased BP subsequent to an i.v. injection of the CB_1_R antagonist AM251. In the present experiments, CP55940 increased SBP, DBP and MBP by ~8–15 mmHg in WKY (Figure 6a) and by ~15–18 mmHg in SHR (Figure 6b); the difference was statistically significant for SBP. All antagonists under study, losartan (AT_1_R), PD123319 (AT_2_R) and A-779 (MasR) significantly reduced pressor responses to CP55940 both in normotensive (Figure 6a,c) and hypertensive (Figure 6b,d) rats; in SHR, reduction ranged from ~50% (PD123319 for SHR) to ~85% (A-779 for SHR).

### 3.5. Comparison of Cannabinoid CB_1_, Mas, and AT_2_ Receptor Expression in the PVN, RVLM and NTS of WKY and SHR Rats

As shown in Figure 7a, Western blot analysis revealed an about two-fold higher expression of CB_1_Rs in the PVN and RVLM of SHR compared to their normotensive controls (WKY); there were no differences in NTS. By contrast, the density of AT_2_ and Mas receptors was altered in an opposite direction; this held also true for NTS (Figure 7b,c). AT_2_R expression was about 2–4-fold and Mas receptor expression about 1.5–2-fold higher for WKY than for SHR.

## 4. Discussion

In the present study, we identified interactions between cannabinoid CB_1_Rs and AT_1/2_ receptors for Ang II and MasRs for Ang 1–7 in the PVN involved in blood pressure regulation. The experiments were performed on conscious rats and revealed quantitative differences between SHR and WKY. Conscious rats were used since anesthesia reverses BP response to cannabinoids (e.g., THC, AEA or CP55940) relative to prolonged hypotension (for review, see [17,18]). We used the CB_1/2_ receptor agonist CP55940 [26] since the effect of this drug microinjected into PVN was reduced by the microinjection of the CB_1_R antagonist AM251 but not CB_2_R antagonist SR144528 in PVN [19]. We restricted ourselves to pressor responses since the fall in BP elicited by microinjection of CP55940 into PVN was only reduced by AM251 but not by the antagonists of other receptors including AT_1_Rs in our previous experiments [20]. CP55940 was routinely examined after a previous i.v. administration of AM251 prevented hypotensive responses to CP55940 [19,20].

Ang II (0.3 nmol/rat), Ang 1–7 (0.3 nmol/rat) and CP55940 (0.1 nmol/rat; after i.v. administration of AM251) microinjected into PVN increased SBP, DBP and MBP in conscious WKY and SHR. The strongest pressor response was elicited by Ang II and the weakest one by Ang 1–7 (by ~20 and ~6 mmHg in WKY, respectively). Sun et al. [11] showed that the same doses of Ang II and Ang 1–7, given in PVN, induced comparable increases in MBP in conscious male Sprague-Dawley rats. CP55940 given in PVN (after i.v. administration of AM251) increased BP both in conscious (current study) and urethane-anaesthetized rats [19,20]. By contrast, as mentioned above, the i.v. administration of cannabinoids mainly increased BP in conscious animals but decreased it in anaesthetized animals [17], suggesting that central mechanisms are responsible for the pressor effects of cannabinoids.

The pressor responses to Ang II and Ang 1–7 result from the activation of AT_1_/AT_2_ receptors and Mas receptors, since they were almost prevented or strongly inhibited by respective antagonists losartan/PD123319 and A-779 (given into the PVN). AT_1_Rs in the PVN are well known to mediate the pressor response to Ang II. By contrast, central AT_2_Rs are rather known to counteract the effects of Ang II (for review, see [27]). As in our study, not only losartan but also PD123319 strongly reduced Ang II-induced increases in BP in the PVN of anaesthetized rats [28,29]. Sun et al. [11] showed that losartan inhibited the pressor effects induced by Ang II but not by Ang 1–7, which were reduced by A-779, confirming the involvement of AT_1_- and MasRs in the increases in BP elicited by Ang II and Ang 1–7, respectively. As mentioned above, the involvement of CB_1_Rs in the pressor effect of CP55940 after its administration into PVN has been proven in our previous study [19].

To the best of our knowledge, we are the first to demonstrate that the pressor responses to Ang II, Ang 1–7 and CP55940 were significantly higher (or tended to be higher) in SHR than in WKY. Higher increases in BP induced by Ang II [10,11] and Ang 1–7 [11,15] have so far been shown in renovascular hypertensive rats only. Increases/decreases in blood pressure in response to a given ligand depend on the value of basal BP. Thus, we cannot exclude the possibility that the higher responses observed in SHR were related to higher basal BP. However, we would like to underline that the higher pressor effects in SHR, when compared to WKY, may result from different densities of the particular receptors in the brain regions responsible for cardiovascular system regulation. The balance of excitatory and inhibitory synaptic inputs depending on glutamatergic and GABAergic transmission is responsible for the final integration of the sympathetic outflow by PVN [1,4,27]. In PVN, facilitatory AT_1_Rs and AT_2_Rs are expressed primarily on glutamatergic and GABAergic neurons, respectively. In addition, AT_1_Rs diminish GABAergic inputs [13,27,30,31]. Presynaptic inhibitory CB_1_Rs are localized both on glutamatergic and GABAergic synapses [32]. However, we have provided evidence previously that CP55940 increased BP, acting predominantly at inhibitory presynaptic CB_1_Rs on GABAergic neurons [20]. Both an enhancement of sympathoexcitatory and a reduction in sympathoinhibitory inputs might lead to increases in BP.

Compared to WKY, the PVN of SHR has higher levels of AT_1_ [12,13] and CB_1_ (current study) receptors. AT_1_ and CB_1_ receptor expressions are also higher in SHR than WKY in other brain regions, that are crucial for cardiovascular regulation, such as in RVLM and NTS, in which similarly to PVN, glutamate and GABAs interact to modulate BP (for AT_1_Rs—[12,33]; for CB_1_Rs—current study). A higher level of CB_1_Rs but not of endocannabinoids in the RVLM of SHR has been described previously by Wang et al. [34]. To summarize, enhanced AT_1_R (stimulation of glutamatergic neurons) and CB_1_R (inhibition of GABAergic neurons) densities in the PVN of SHR might explain higher BP response to Ang II and CP55940 in SHR than in WKY (Figure 8).

With respect to AT_2_ and Mas receptors, Western blotting analysis showed significantly higher expression in WKY than SHR in each of the three brain regions: PVN, RVLM and NTS. AT_2_ and Mas receptors are known to counteract the pressor effect of AT_1_R stimulation by the enhancement of GABAergic input directly by a facilitatory effect on GABAergic neurotransmission (AT_2_Rs are located predominantly on GABAergic neurons; [27]) or indirectly via nitric oxide stimulating GABA release determined in the PVN both for AT_2_ [35] and Mas [36] receptors. Thus, the decrease in AT_2_ and Mas receptor expression and of the sympathoinhibitory input observed in our study might contribute to the higher level of BP in SHR. Of course, one should keep in mind that a change in receptor expression does not always correlate to functionality, and our results require confirmation by studies dedicated to the downstream pathways regarding specific receptor activation. Interestingly, the density of MasRs in the PVN was higher in renal hypertensive rats (2K1C) compared to their normotensive controls [15]. However, the following question arises: how can we explain the pressor effects of Ang 1–7 and the involvement of AT_2_Rs in the increase in BP induced by Ang II? The enhancement of BP elicited by Ang 1–7 has so far been explained by the production of reactive oxygen species (ROS; [16]) and the pressor response induced by AT_2_Rs is probably mediated via vasopressinergic neurons in the PVN [37], since the increases in BP and HR induced by vasopressin microinjected into the PVN of anaesthetized rats were partially attenuated by the AT_1_R antagonist losartan and strongly reduced by the AT_2_R antagonist PD123319 [37].

The following facts confirm the existence of an interaction between cannabinoid CB_1_Rs, AT_1_ and AT_2_ receptors for Ang II and MasRs for Ang 1–7 in the PVN. Firstly, the CB_1_R antagonist AM251 given into the PVN reduced or tended to diminish the pressor response to Ang II both in WKY and SHR and even reversed increases in BP elicited by Ang 1–7 to hypotensive responses. An inhibitory effect of AM251 on the pressor response to Ang II (both given into the PVN) was previously shown by Gyombolai et al. [21]. Secondly, the pressor response to CP55940 was strongly reduced by previous topical applications of AT_1_, AT_2_ and Mas receptor antagonists losartan, PD123319 and A-779, respectively. In our previous papers, losartan i.v. did not modify the cardiovascular response to AEA i.v. (including its pressor phase; [38]) and the hypotensive and bradycardic responses to CP55940 microinjected into PVN. On the other hand, it even reversed the increase in BP and HR that was induced by this latter compound injected in PVN [20]. In the current study, losartan did not reverse the pressor response to CP55940 in BP probably because it was given locally to PVN in contrast to i.v. injection in the previous paper [20]. Thirdly, topically administered A-779 decreased BP by itself to a comparable degree both in WKY and SHR but only after the previous i.v. administration of AM251. By contrast, losartan and PD123319 failed to modify BP both in the absence or presence of AM251 i.v. Thus, it seems that endocannabinoids acting on CB_1_ receptors in PVN interact with the endogenous pressor tone elicited by Ang 1–7 but not Ang II. Similarly, it has been shown previously that A-779 (3 nmol/rat) given in PVN reduced BP, renal sympathetic nerve activity and plasma noradrenaline levels in anaesthetized high salt-induced or renal hypertensive rats but not in the respective control animals [11,16]. We are the first to demonstrate an interaction of Ang 1–7 and its MasRs with cannabinoid CB_1_Rs in the central nervous system. Thus far, the existence of such an interaction was suggested only in the rat uterus in which local Ang 1–7 infusion enhanced CB_1_ and CB_2_ receptor expression [22].

How can we explain the interaction between CB_1_ receptors with AT_1_, AT_2_ and Mas receptors (Figure 8)? The paracrine transactivation of CB_1_ receptors by endocannabinoids released after activation of AT_1_Rs by Ang II in cells expressing both AT_1_ and CB_1_ receptors diminished the response to Ang II in various cultured cells [39] and vessels (for review, see [40]) and in the heart [41]. On the basis of the latter data, one might assume that AT_1_R activation in PVN leads to the release of endocannabinoids, which in turn activate presynaptic inhibitory CB_1_Rs mainly on GABAergic neurons (for detailed explanation, see above). Thus, the CB_1_R antagonist AM251 inhibits the pressor response to Ang II since it antagonizes presynaptic CB_1_Rs responsible for the reduction in GABA release. Consequently, the enhanced inhibitory influence on glutamatergic input reduces the pressor response to Ang II. On the other hand, in our hands, losartan inhibited the pressor response to CP55940, which mainly results from the reduction in inhibitory GABAergic tones via activating presynaptic CB_1_Rs by their agonist (see above). Since the final BP response in PVN is determined by the balance between GABAergic and glutamatergic inputs, we suppose that losartan blocking AT_1_Rs on glutamatergic neurons might reduce the pressor response to CP55940. We are not able to comment on the mechanisms related to AT_2_ and Mas receptors because only very limited data regarding their pressor effects are available so far. One should keep in mind that impaired GABAergic and/or enhanced glutamatergic inputs result in elevated sympathetic outflow and hypertension (including SHR) [4]. In our hands, AM251 reduced the pressor response to Ang II in SHR more strongly than in WKY, confirming the above suggestion that it mainly inhibits presynaptic inhibitory CB_1_ receptors located mainly on GABAergic neurons.

Interestingly, CB_1_R antagonists enhanced vasoconstrictor responses to Ang II in various isolated arteries (for review, see [40]) but decreased the increase in BP induced by the microinjection of Ang II into the PVN (current paper), confirming that CB_1_R activation in the peripheral vascular bed and in the PVN is mainly responsible for hypotensive and hypertensive effects of endocannabinoids, respectively (for review, see [18]).

In this context, it is also of interest to comment on the effect of the CB_1_R antagonist AM251 when given alone. AM251 administered i.v. and into PVN increased BP by about 15–20 mmHg and 6–8 mmHg, respectively, both in WKY and SHR. Its effect following i.v. injection probably results from the activation of peripheral presynaptic inhibitory CB_1_Rs located on sympathetic nerve endings innervating resistance vessels [42]. Accordingly, the fall in BP in response to CP55940 in anaesthetized rats was reversed to hypertension by the peripheral CB_1_R antagonist AM6545 [19]. On the other hand, the weaker enhancement of BP after its application in PVN might be caused by the blockade of CB_1_Rs responsible for the fall in BP by endocannabinoids released in the PVN (first phase of the response to CP55940; [19,20]. An increase in BP by AM251 given in PVN was also observed in the urethane-anaesthetized [21] (0.3 nmol/rat) rats but not by our group [19]. In our previous study, AM251 given i.v. also failed to affect BP in urethane-anaesthetized rats; they have a lower sympathetic tone (lower basal BP than in the present paper; [19,20]) and probably also a lower activity with respect to the endocannabinoid system.

## 5. Conclusions

We show that microinjections of Ang II, Ang 1–7 and CP55940 (after i.v. administration of the CB_1_ receptor antagonist AM251) in PVN induced pressor responses in conscious rats via AT_1/2_, Mas and CB_1_ receptors, respectively, and that the pressor responses were stronger in SHR than in WKY. The more pronounced pressor effect in SHR may partially result from its higher (AT_1_ and CB_1_) and lower (AT_2_ and Mas) receptor expressions in PVN, RVLM and NTS compared to WKY. With respect to PVN, we also showed a mutual interaction between cannabinoid CB_1_ receptors and receptors for Ang II and Ang 1–7 responsible for the stimulation of the pressor response. These interactions, the mechanisms of which have to be clarified in future studies, have to be considered when compounds acting at CB_1_ and AT_1_ receptors are used for therapeutic purposes.

## Figures and Tables

**Figure 1 cells-11-01542-f001:**
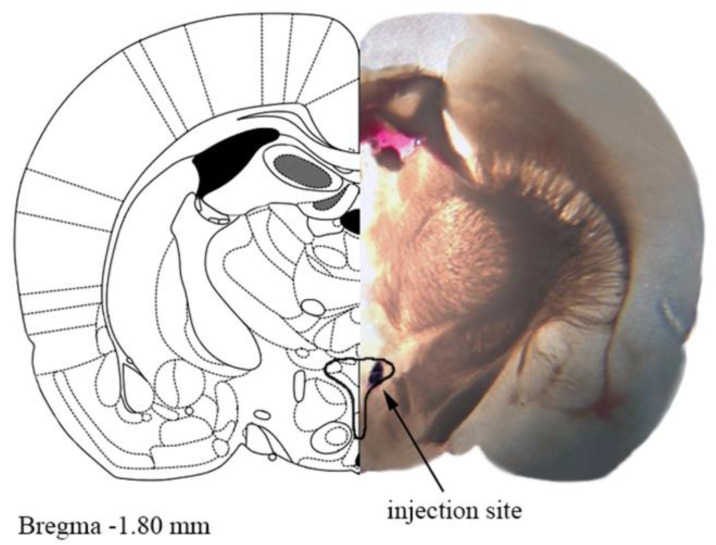
A representative photograph of the microinjection site in the paraventricular nucleus of hypothalamus (PVN) coupled with the matching slide from the Paxinos and Watson rat brain atlas [24]. One mm thick brain slice with the injection site shown by Evans blue dye. The drawn outline represents the confines of PVN.

**Figure 2 cells-11-01542-f002:**
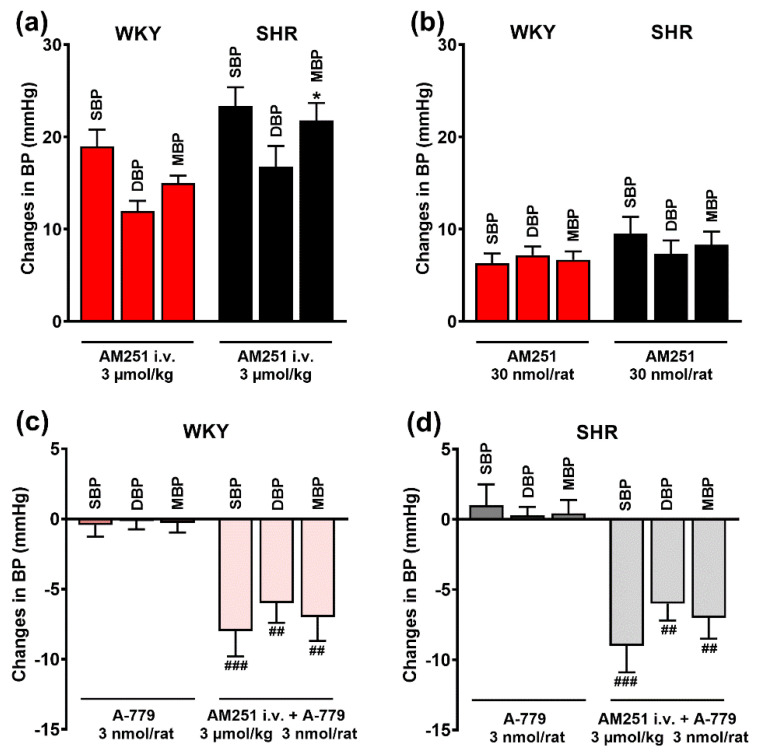
Effects of AM251 and A-779 on basal systolic (SBP), diastolic (DBP) and mean (MBP) blood pressure in conscious Wistar Kyoto (WKY) and spontaneously hypertensive rats (SHR). CB_1_ receptor antagonist AM251 was injected intravenously (**a**) or into the paraventricular nucleus of hypothalamus (PVN) (**b**). Mas receptor antagonist A-779 was injected into the PVN. Results are presented as mean ± SEM (*n* = 21–27 for A; *n* = 10 for B; *n* = 6–7 for (**c**) and (**d**). * *p* < 0.05 compared to WKY; ## *p* < 0.01, ### *p* <0.001 compared to A-779 in the absence of AM251.

**Figure 3 cells-11-01542-f003:**
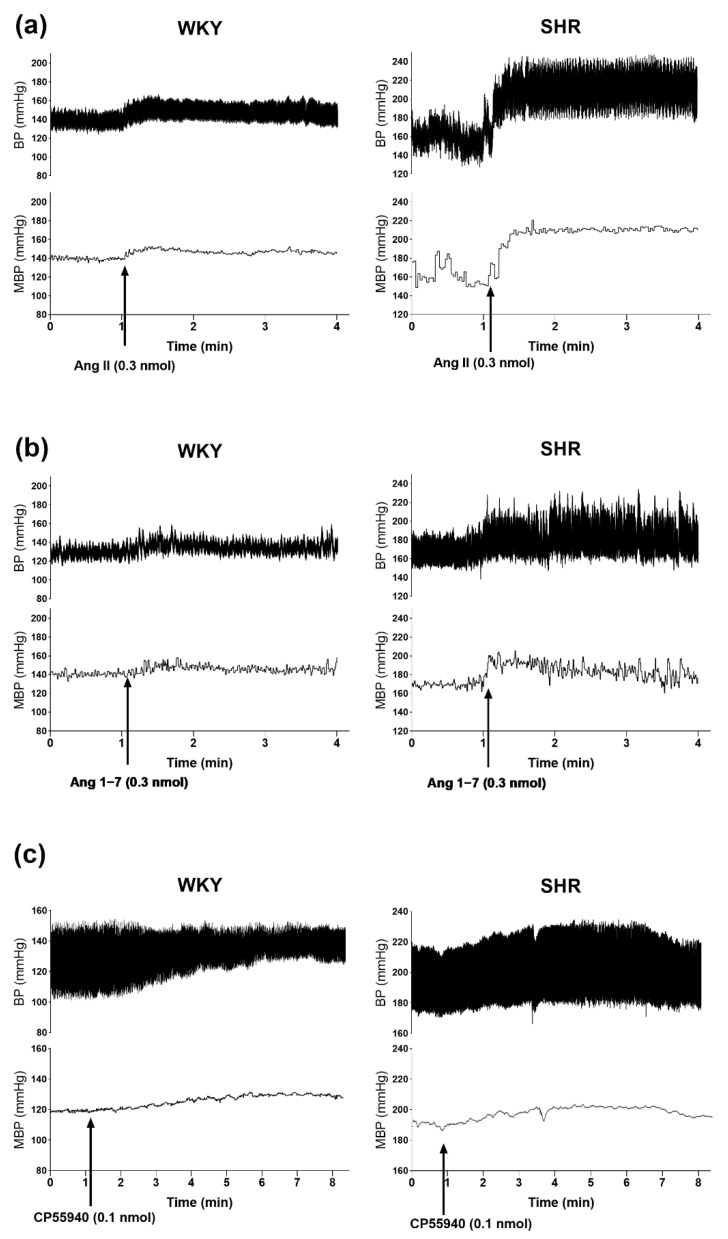
Representative original traces of the pressor effects of angiotensin II (**a**), angiotensin 1–7 (**b**) and CP55940 (after i.v. administration of AM251) (**c**) injected into the paraventricular nucleus of hypothalamus (PVN) on blood pressure (BP) and mean blood pressure (MBP) in conscious Wistar Kyoto (WKY) and spontaneously in hypertensive rats (SHR). Arrows show the moment of application of the particular agonist.

**Figure 4 cells-11-01542-f004:**
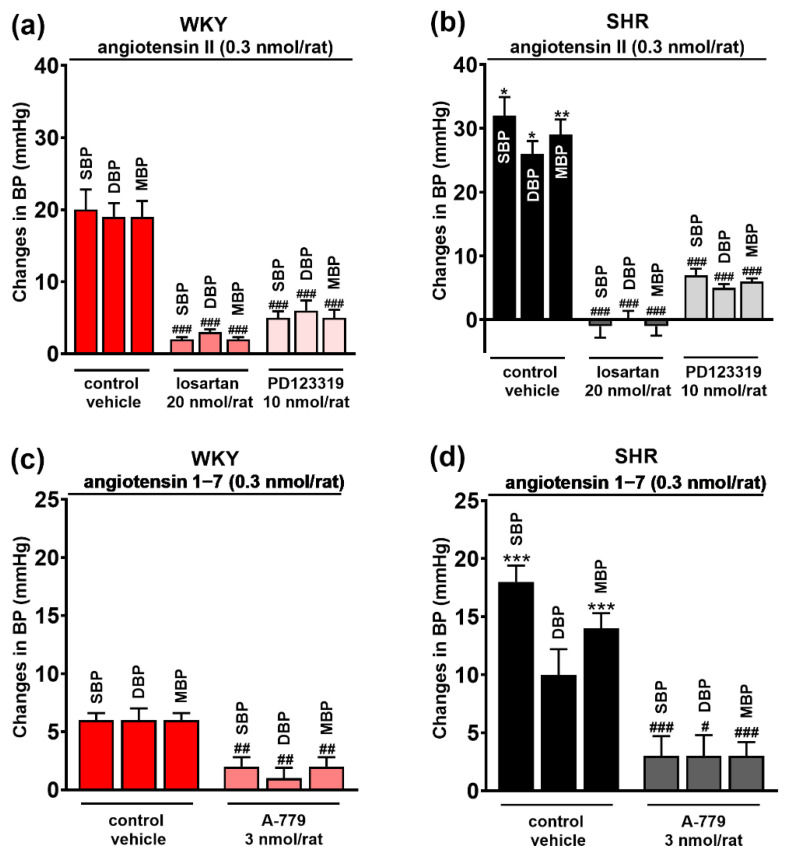
Effect of angiotensin II (AT_1/2_ receptor agonist) (**a**,**b**) and angiotensin 1–7 (Mas receptor agonist) (**c**,**d**) on systolic (SBP), diastolic (DBP) and mean (MBP) blood pressure, and their interaction with AT_1_, AT_2_ and Mas receptor antagonists (losartan, PD123319 and A-779, respectively) injected into the paraventricular nucleus of hypothalamus (PVN) in conscious Wistar Kyoto (WKY) and spontaneously hypertensive rats (SHR). Results are presented as mean ± SEM (*n* = 7–8) and calculated as change in basal values determined before agonist injection. # *p* < 0.05, ## *p* < 0.01, ### *p* < 0.001 compared to the corresponding control group; * *p* < 0.05, ** *p* < 0.01, *** *p* < 0.001 compared to WKY.

**Figure 5 cells-11-01542-f005:**
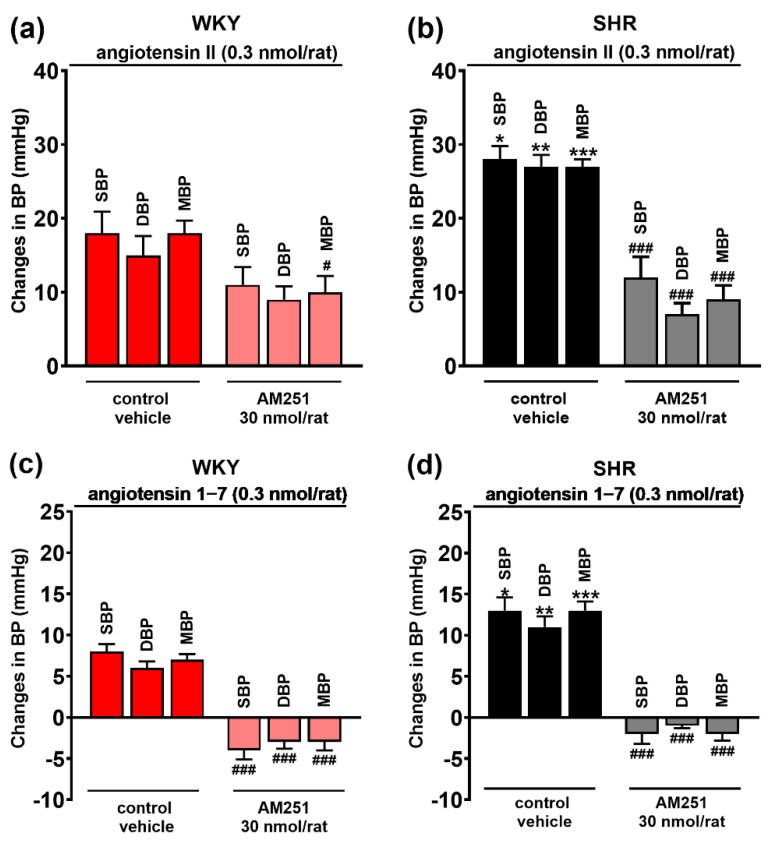
Influence of the CB_1_ receptor antagonist AM251 on the pressor effect of angiotensin II (AT_1/2_ receptor agonist) (**a**,**b**) and angiotensin 1–7 (Mas receptor agonist) (**c**,**d**) injected into the paraventricular nucleus of hypothalamus (PVN) in conscious Wistar Kyoto (WKY) and spontaneously hypertensive rats (SHR). Results, presented as mean ± SEM (*n* = 7–8), are calculated as change in basal values determined before agonist/antagonist injection. # *p* < 0.05, ### *p* < 0.001 compared to the corresponding control group. * *p* < 0.05, ** *p* < 0.01, *** *p* < 0.001 compared to WKY.

**Figure 6 cells-11-01542-f006:**
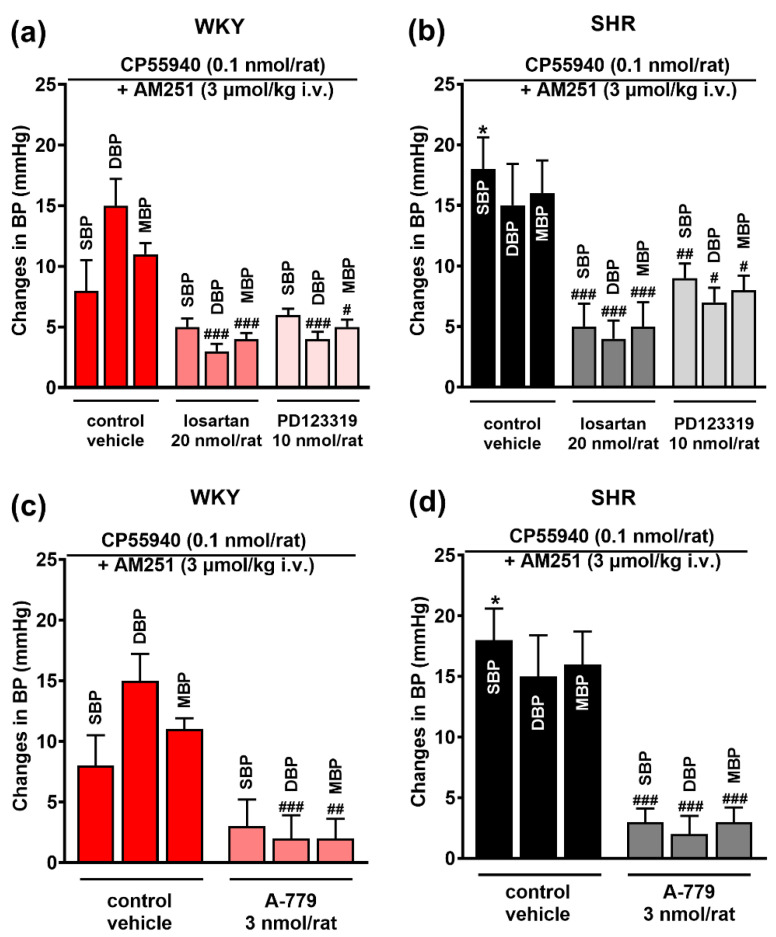
Influence of AT_1_ and AT_2_ (losartan and PD123319; (**a**,**b**), and Mas (A-779; (**c**,**d**) receptor antagonists on the pressor effect of the CB_1_ receptor agonist CP55940 injected into the paraventricular nucleus of hypothalamus (PVN) in conscious Wistar Kyoto (WKY) and spontaneously hypertensive rats (SHR). All experiments were performed after i.v. injection of the CB_1_ receptor antagonist AM251. Results, presented as mean ± SEM (*n* = 5–7), are calculated as the change in basal values determined before agonist injection. *# p* < 0.05, #*# p* < 0.01, ##*# p* < 0.001 compared to the corresponding control group; ** p* < 0.05 compared to WKY.

**Figure 7 cells-11-01542-f007:**
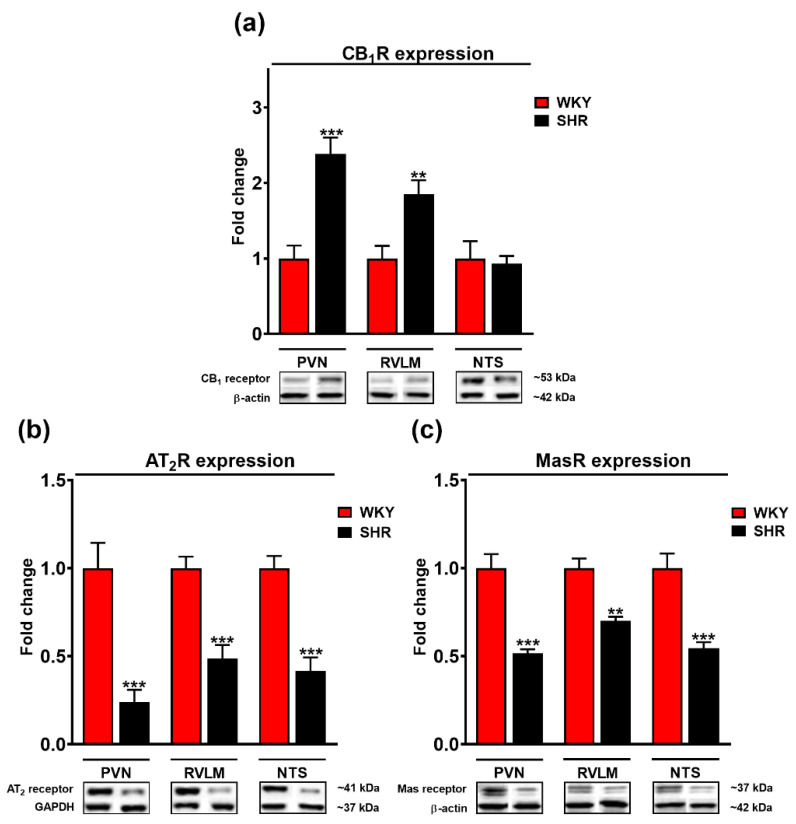
Fold change of cannabinoid CB_1_ (**a**), AT_2_ (**b**) and Mas (**c**) receptors and representative Western blots of the paraventricular nucleus of hypothalamus (PVN), rostral ventrolateral medulla (RVLM) and nucleus tractus solitarii (NTS) in Wistar Kyoto (WKY) and spontaneously hypertensive rats (SHR). β-actin or glyceraldehyde-3-phosphate dehydrogenase (GAPDH) served as loading control. Results presented as mean ± SEM (*n* = 6). ** *p* < 0.01, *** *p* < 0.001 compared to WKY. Uncropped Western blot images can be seen in Appendix A.

**Figure 8 cells-11-01542-f008:**
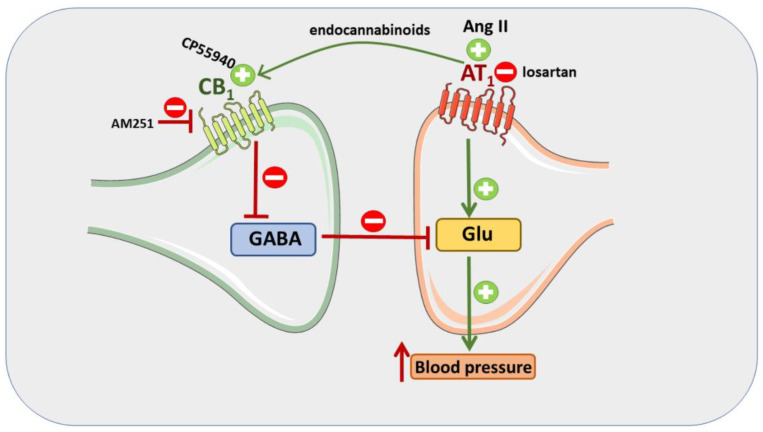
Mechanisms involved in the pressor response induced by Ang II and CP55940. Ang II, via AT_1_ receptors, leads to endocannabinoid formation in the paraventricular nucleus of hypothalamus (PVN). Endocannabinoids in turn inhibit GABA release by activation of CB_1_ receptors. Since little GABA is released, much glutamate (Glu) can be released, also since facilitatory AT_1_ receptors on the glutamatergic neuron further increase Glu release. The extents of Glu release and of the pressor response are correlated. The pressor response in SHR increased since AT_1_ and CB_1_ receptor density in their PVN increased. On the other hand, the pressor response induced by Ang II can be attenuated by CB_1_ receptor antagonist AM251. In addition, the pressor response induced by the CB_1_ receptor agonist CP55940 can be antagonized by AT_1_ receptor antagonist losartan.

## Data Availability

Not applicable.

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
