# Peer review of "Cross-Talk between CB1, AT1, AT2 and Mas Receptors Responsible for Blood Pressure Control in the Paraventricular Nucleus of Hypothalamus in Conscious Spontaneously Hypertensive Rats and Their Normotensive Controls"

_cells, 2022, doi:10.3390/cells11091542_

Round 1
Reviewer 1 Report
The present manuscript by Mińczuk et al. evaluate the interactions between cannabinoid CB1Rs with AT1 and AT2 receptors for Ang II and Mas receptors for Ang 1-7 in the PVN of WKY and SHR conscious rats and their involvement in blood pressure regulation. The present findings are of interest since the authors are the first to demonstrate that the pressor responses to Ang II, Ang 1-7 and CP55940 were significantly higher (or tended to be higher) in SHR than in WKY. The methodology used is consistent with the aim of the work. The results are clear, and the discussion is complete and well-founded.
Some remarks might be interesting to address (in text or preferentially by experiment):
- The authors hypothesized that the higher pressor effects in SHR, when compared to WKY, may result from different densities of the receptors in the brain regions responsible for cardiovascular system regulation. It would be of interest to evaluate receptor activation (e.g., evaluating a signaling pathway activated by the agonist) involved in the pressor response, since a change in receptor expression does not always correlate to functionality.
- Since the authors showed the existence of a mutual interaction between cannabinoid CB1 receptors and receptors for Ang II and Ang 1-7 by using selective antagonism with AT1 (losartan), AT2 (PD123319) and CB1 (AM251) receptors blockers, it would also be of interest to show the existence of a physical interaction between CB1Rs with AT1 and AT2 and Mas receptors. This interaction could be shown by using the technique of co-immunoprecipitation.
- As cited by the authors (page 12, lines 342 and 343) central AT2Rs are known to counteract the effects of Ang II. In this sense, the authors give little explanation about the involvement of AT2Rs in the increase in BP induced by Ang II that they observed in their results. It would be interesting if the authors could expand their explanation about this result.
- Figure 7 shows the expression of CB1 (a), AT2 (b) and Mas (c) receptors. AT1 receptor expression was not evaluated?
- Regarding the multiple bands that appear in the images corresponding to the western blots of the receptors, have they been reported by the manufacturer or by other authors? What could they be due to?
- It would be interesting to add a final sentence about future perspectives and the implication of the study findings on the knowledge of the subject in question.
Author Response
Dear Sir or Madam,
In your referee report, you have written:
The present manuscript by Mińczuk et al. evaluate the interactions between cannabinoid CB1Rs with AT1 and AT2 receptors for Ang II and Mas receptors for Ang 1-7 in the PVN of WKY and SHR conscious rats and their involvement in blood pressure regulation. The present findings are of interest since the authors are the first to demonstrate that the pressor responses to Ang II, Ang 1-7 and CP55940 were significantly higher (or tended to be higher) in SHR than in WKY. The methodology used is consistent with the aim of the work. The results are clear, and the discussion is complete and well-founded.
Thank you for your compliments.
Some remarks might be interesting to address (in text or preferentially by experiment):
- The authors hypothesized that the higher pressor effects in SHR, when compared to WKY, may result from different densities of the receptors in the brain regions responsible for cardiovascular system regulation. It would be of interest to evaluate receptor activation (e.g., evaluating a signaling pathway activated by the agonist) involved in the pressor response, since a change in receptor expression does not always correlate to functionality.
We completely agree with you that a change in receptor expression does not always correlate to functionality. Our results were consistent with our hypothesis, but additional studies with focus on the downstream pathways regarding specific receptor activation are necessary. We have added a sentence to the Discussion (page 15, lines 486-488). The revised parts of our manuscript are marked with yellow colour.
- Since the authors showed the existence of a mutual interaction between cannabinoid CB1 receptors and receptors for Ang II and Ang 1-7 by using selective antagonism with AT1 (losartan), AT2 (PD123319) and CB1 (AM251) receptors blockers, it would also be of interest to show the existence of a physical interaction between CB1Rs with AT1 and AT2 and Mas receptors. This interaction could be shown by using the technique of co-immunoprecipitation.
Co-immunoprecipitation is in fact a great technique to show physiologically-relevant protein-protein interactions. We were not able to perform additional examinations because of limited tissue availability. We decided to focus on Western blots, but the co-IP method should be used in a future project connected with new functional experiments.
- As cited by the authors (page 12, lines 342 and 343) central AT2Rs are known to counteract the effects of Ang II. In this sense, the authors give little explanation about the involvement of AT2Rs in the increase in BP induced by Ang II that they observed in their results. It would be interesting if the authors could expand their explanation about this result.
We have extended our previous explanation (page 12, lines 395-400).
- Figure 7 shows the expression of CB1 (a), AT2 (b) and Mas (c) receptors. AT1 receptor expression was not evaluated?
The expression of AT1 receptors in WKY and SHR PVN, RVLM and NTS was well established by other researchers, as we mentioned in our manuscript [12] and [13].
- Regarding the multiple bands that appear in the images corresponding to the western blots of the receptors, have they been reported by the manufacturer or by other authors? What could they be due to?
Thank you for your question. Yes, we realize that there are some unspecific bands in our Western blot pictures. CB1R, AT2R, and MasR are not always easy receptors to blot. Unspecific bands are quite common.
You can see non-specific binding while using specific antibodies in the following publications:
AT2 receptors
Gao J, Chao J, Parbhu KJ, et al. Ontogeny of angiotensin type 2 and type 1 receptor expression in mice. J Renin Angiotensin Aldosterone Syst. 2012;13(3):341-352. doi:10.1177/1470320312443720
Franco R, Lillo A, Rivas-Santisteban R, et al. Functional Complexes of Angiotensin-Converting Enzyme 2 and Renin-Angiotensin System Receptors: Expression in Adult but Not Fetal Lung Tissue. Int J Mol Sci. 2020;21(24):9602. Published 2020 Dec 16. doi:10.3390/ijms21249602. Specifically in Supplementary Figure S1, although only a small portion of the original blot is shown.
Ewert, S., Laesser, M., Johansson, B., Holm, M., Aneman, A., & Fandriks, L. (2003). The angiotensin II receptor type 2 agonist CGP 42112A stimulates NO production in the porcine jejunal mucosa. BMC pharmacology, 3, 2. https://doi.org/10.1186/1471-2210-3-2. Again, you can see multiple bands, even though only a small portion of the gel is shown.
Mas receptors:
Burghi, V., Fernández, N. C., Gándola, Y. B., Piazza, V. G., Quiroga, D. T., Guilhen Mario, É., Felix Braga, J., Bader, M., Santos, R., Dominici, F. P., & Muñoz, M. C. (2017). Validation of commercial Mas receptor antibodies for utilization in Western Blotting, immunofluorescence and immunohistochemistry studies. PloS one, 12(8), e0183278. https://doi.org/10.1371/journal.pone.0183278
Yu, X., Cui, L., Hou, F., Liu, X., Wang, Y., Wen, Y., Chi, C., Li, C., Liu, R., & Yin, C. (2018). Angiotensin-converting enzyme 2-angiotensin (1-7)-Mas axis prevents pancreatic acinar cell inflammatory response via inhibition of the p38 mitogen-activated protein kinase/nuclear factor-κB pathway. International journal of molecular medicine, 41(1), 409–420. https://doi.org/10.3892/ijmm.2017.3252
CB1 receptors:
Rapino, C., Castellucci, A., Lizzi, A. R., Sabatucci, A., Angelucci, C. B., Tortolani, D., Rossi, G., D'Andrea, G., & Maccarrone, M. (2019). Modulation of Endocannabinoid-Binding Receptors in Human Neuroblastoma Cells by Tunicamycin. Molecules (Basel, Switzerland), 24(7), 1432. https://doi.org/10.3390/molecules24071432
Tyagi, V., Philips, B. J., Su, R., Smaldone, M. C., Erickson, V. L., Chancellor, M. B., Yoshimura, N., & Tyagi, P. (2009). Differential expression of functional cannabinoid receptors in human bladder detrusor and urothelium. The Journal of urology, 181(4), 1932–1938. https://doi.org/10.1016/j.juro.2008.11.078
Unspecific binding is common while using polyclonal antibodies, such as anti-MasR used in our study, since polyclonal antibodies may recognize multiple bands of different molecular weight, due to their ability to recognize multiple epitopes. Fortunately, it does not change the results, since each blot is accompanied with a standard ladder, allowing to accurately identify the molecule of interest.
- It would be interesting to add a final sentence about future perspectives and the implication of the study findings on the knowledge of the subject in question.
We have re-phrased the final sentence or our conclusion.
We believe that we have addressed all the comments of the reviewer. We hope that you now will find our revised manuscript suitable for publication in the Special Issue of Cells “Novel Insights into Cannabinoid Receptors, Molecular Targets, and Therapeutic Potentials”.
Sincerely Yours,
Prof. Dr. Barbara Malinowska
also on behalf of my co-authors
Reviewer 2 Report
This is a very interesting study. The authors report results that explain the pathophysiological background of central regulation of blood pressure by CB1, AT1, AT2 and MasR receptors.
The introduction and discussion are well written, they are easy to understand, and the results compare well with other studies.
The graphical results are clearly described, but there is too much information in the text and it is not very clear for a first reading. It could be improved a bit.
The results are very attractive and I recommend this study for publication.
Author Response
Dear Sir or Madam,
In your referee report, you have written:
This is a very interesting study. The authors report results that explain the pathophysiological background of central regulation of blood pressure by CB1, AT1, AT2 and MasR receptors.
The introduction and discussion are well written, they are easy to understand, and the results compare well with other studies.
The graphical results are clearly described, but there is too much information in the text and it is not very clear for a first reading. It could be improved a bit.
The results are very attractive and I recommend this study for publication.
We would like to thank you for having read our manuscript and for your compliments.
We have improved the description of the results. p values, which are available from the figures, have been deleted from the text five times. The revised parts of our manuscript are marked with yellow colour.
We believe that we have addressed all the comments of the reviewer. We hope that you now will find our revised manuscript suitable for publication in the Special Issue of Cells “Novel Insights into Cannabinoid Receptors, Molecular Targets, and Therapeutic Potentials”.
Sincerely Yours,
Prof. Dr. Barbara Malinowska
also on behalf of my co-authors